# Noise-Canceling Channel Estimation Schemes Based on the CIR Length Estimation for IEEE 802.11p/OFDM Systems

Kyunbyoung Ko [1] and Hanho Wang [2,*]

1    Department of Electronics Engineering, Korea National University of Transportation, 50 Daehak-ro, Chungju-si 27469, Chungbuk, Republic of Korea; kbko@ut.ac.kr
2    Department of Smart Information and Telecommunication Engineering, Sangmyung University, 31, Sangmyeongdae-gil, Dongnam-gu, Cheonan-si 31066, Chungnam, Republic of Korea
*    Correspondence: hhwang@smu.ac.kr

**Abstract:** This paper investigates methods for noise-canceling channel estimation (NC-CE) to track rapid time-varying channels in IEEE 802.11p/orthogonal frequency division multiplexing (OFDM) systems. To this end, we introduce a novel three-step channel estimation technique based on the estimated length of the channel impulse response (CIR). This approach aims to surpass the performance of conventional designs that rely on constructed data pilots (CDPs). In the first step, we not only eliminate noise components but also estimate the channel frequency responses (CFRs) of virtual subcarriers for long preamble parts. Moving on to the second step, we incorporate a modified CDP method without a frequency-domain reliability test and interpolation, taking into account the CFRs of virtual subcarriers obtained at the previous OFDM symbol time. The final step can be implemented as the operation of the inverse fast Fourier transform (IFFT)/nulling/FFT to reduce noise components from the CFRs obtained in the second step and generate CFRs for virtual subcarriers to be used in the next symbol time. The results of our simulations validate the effectiveness of our proposed channel estimation schemes.

**Keywords:** IEEE 802.11p; NC; CE; CIR; CFR



## 1. Introduction

Extensive research has been conducted in the field of cooperative intelligent transportation systems (C-ITS), which aims to actively respond to real-time traffic conditions by facilitating communication between vehicles and infrastructure while in motion [1]. Various candidate technologies have emerged to realize vehicle-to-everything (V2X) communications, and the IEEE 802.11p standard has been developed based on the physical layer (PHY) and medium access control (MAC) layer of WiFi [1,2]. It is worth noting that the IEEE 802.11p standard is essentially a modification of the frequency bandwidth specified in the IEEE 802.11a standard, reducing it from 20 MHz to 10 MHz [1].

The IEEE 802.11p standard has the technical limitations of existing WiFi although it is based on a proven WiFi technology. As is well known, IEEE 802.11a is a technology designed without considering the fast mobility of terminals [1,2]. Accordingly, in the transmission frame of the IEEE 802.11p, each orthogonal frequency division multiplexing (OFDM) symbol includes only four pilot subcarriers [1]. Research on channel estimation methods has been extensively performed for OFDM systems [2–10]. Nevertheless, with this small number of pilot subcarriers, it is difficult to estimate with high precision channel changes in the frequency domain that occur in V2X situations [3–7]. In particular, considering that information exchange through V2X communication is very essential for vehicle driving safety, it is necessary to emphasize more accurate channel estimation within the constraints on the pilot subcarrier of IEEE 802.11p [2,8–12].

Using only a limited number of pilot signals, research has continued to improve channel estimation performance for IEEE 802.11p/OFDM systems [3–5,11–15]. To overcome

the lack of a pilot signal, the construct data pilot (CDP) method is proposed [10]. In this CDP method, the data symbols detected by both the estimated channel frequency response (CFR) from the current four pilot symbols and the estimated CFR at the previous OFDM symbol are used as constructed pilot signals to improve channel estimation performance. The channel estimation technique using the CDP can further improve channel estimation performance by introducing the time-domain reliability test and frequency-domain interpolation (TRFI) [4,5,11]. Virtual carriers are utilized for channel estimation based on the CDP using the TRFI, which can improve the channel estimation performance in IEEE 802.11p, as shown in previous studies [4,5].

Let us take a closer look at the channel estimation technique that shows the best performance in this research area [3–5]. In [3], the authors found that much noise remained in the channel estimated in the frequency domain using pilot subcarriers. To reduce this noise, they first obtain the time-domain channel impulse response (CIR) using the inverse discrete Fourier transform (IDFT) on the estimated CFR. Next, the noise present in the estimated channel can be reduced by forcing the values to zero that exist after the maximum delay spread in the obtained CIR. Although this channel estimation technique shows robust performance against noise, the authors provided the performance of this channel estimator assuming that the maximum delay spread of the CIR is accurately known. In the real world, the maximum delay spread of the CIR cannot be accurately known, and only an approximate value can be known through estimation. Notice that in the studies by the authors of [4,5], the assumption was made that when estimating the CFR of virtual subcarriers, the CIR length is arbitrarily chosen as $L_{\mathrm{cp}} - 1$, where $L_{\mathrm{cp}}$ is the cyclic prefix length. In [4], the authors improve channel estimation accuracy by replacing the channel leakage energy (CLE) present in the null subcarriers of most practical OFDM systems with the successive interpolation scheme. While existing studies have only attempted to mitigate the CLE presented by null subcarriers [13–15], the approaches proposed in [4,5] involve estimating the CFRs for virtual subcarriers to obtain excellent channel estimation performance. Nevertheless, they require very high computational complexity in the frequency-domain interpolation process of obtaining the CFR for the CLE substitution. Also, the authors in [12,16] deal with the minimum mean square error (MMSE) channel estimation techniques with reduced complexity. In [12,16], it is based on the CIR length estimation and the power delay profile estimation, respectively. Nevertheless, they still require high computational complexity related to the MMSE solution.

In this paper, building upon the methodologies presented in [4,5], we first release the conventional assumption that the maximum delay spread is perfectly known [3] and verify the proposed scheme to be implemented by a practical maximum delay spread determination criterion of [6,7]. Through this, we provide a noise-canceling channel estimation (NC-CE) that can be practically implemented in the time domain for IEEE 802.11p/OFDM channel estimation. Note that the proposed NC-CE method uses only fast Fourier transform (FFT) and inverse FFT to generate channel values for null subcarriers and does not use additional frequency-domain interpolation, thereby significantly reducing computational complexity compared to the existing channel estimation techniques.

The remainder of this paper is organized as follows: Section 2 describes the IEEE 802.11p physical layer. The proposed channel estimation scheme is described in Section 3. Section 4 shows the simulation results, and concluding remarks are given in Section 5.

## 2. System Model

IEEE 802.11p has been standardized by modifying some specifications in the physical layer of the existing IEEE 802.11a [1]. The IEEE 802.11p standard utilizes the 5.9 GHz band and a bandwidth of 10 MHz, which is half of 802.11a bandwidth. The size of the FFT and IFFT is 64 (=$N$). With one OFDM symbol duration equal to $T_s = 6.4$ µs, the transmitter employs the convolutional encoder, and the receiver adopts the Viterbi decoder [1,5].

The IEEE 802.11p packet is composed of three parts: the preamble for time–frequency synchronization, the signal field for control information, and the data field for message

signals [1,12]. The preamble located at the beginning of the packet has short training symbols used for time synchronization and two long training symbols used for the initial channel estimation. Other specific parameters in IEEE 802.11p can be found in [1].

IEEE 802.11p's physical layer is based on OFDM. The guard interval (GI) is arranged to reduce inter-symbol interference (ISI) due to multipath fading channels. The signal field composed of one OFDM symbol has information such as modulation order, code rate, and so on. On the other hand, the data field contains data to be transmitted and the number of OFDM symbols in the data field can be variable according to data size. In the data field, each OFDM symbol contains $N$ (=64) subcarriers, including $N_p$ (=4) pilot subcarriers with an index set of $\mathbb{S}_p = \{-21, -7, 7, 21\}$, $N_v$ (=12) virtual (null) subcarriers with an index set of $\mathbb{S}_v = \{-32, \cdots, -27, 0, 27, \cdots, 31\}$, and $N_d$ (=48) data subcarriers with an index set of $\mathbb{S}_d$. Then, the number of useful subcarriers is $N_u$ (=$N_p + N_d = 52$) with an index set of $\mathbb{S}_u = \mathbb{S}_d \cup \mathbb{S}_p = \{-26, -25, \cdots, -1, 1, \cdots, 25, 26\}$. From here, the diagonal matrices $\mathbf{p}_p$, $\mathbf{p}_v$, $\mathbf{p}_d$, and $\mathbf{p}_u$ (=$\mathbf{p}_p + \mathbf{p}_d$) are defined so that the non-zero diagonal elements of the matrix indicate the positions of pilot subcarriers, virtual subcarriers, data subcarriers, and useful subcarriers, respectively. For example, the $r$th row and $c$th column element of $\mathbf{p}_p$ can be presented as

$$\mathbf{p}_p(r, c) = \begin{cases} 1, & \text{if } r = c \ \& \ r \in \mathbb{S}_p \\ 0, & \text{else.} \end{cases} \tag{1}$$

### 2.1. Notations

Throughout this paper, normal letters represent scalar values, and boldface letters denote vectors or matrices, respectively. The superscripts $(\cdot)^\mathsf{T}$ and $(\cdot)^\mathsf{H}$ stand for the transpose and conjugate transpose of a matrix. Also, diag$(\cdot)$ means to convert an input diagonal matrix to an output column vector or an input column vector to an output diagonal matrix. Note that $\mathbf{A}./\mathbf{B}$ indicates the element-by-element division between the two vectors $\mathbf{A}$ and $\mathbf{B}$.

When we define $\mathbf{F}$ as the $(N \times N)$ full FFT matrix, $\mathbf{F}_{64}$ and $\mathbf{F}_{52}$ represent the reduced $(64 \times L)$ FFT matrix obtained by taking 64 (=$N$) rows and the first $L$ columns of $\mathbf{F}$ and the reduced $(52 \times L)$ FFT matrix obtained by taking 52 (=$N_u$) rows corresponding to the useful subcarriers in $\mathbb{S}_u$ and the first $L$ columns of $\mathbf{F}$, respectively, as

$$\mathbf{F}_{64} = \mathbf{F} \times \begin{bmatrix} \mathbf{I}_{L \times L} \\ \mathbf{0} \end{bmatrix} = \mathbf{F} \times \mathbf{I}_T \tag{2}$$

and

$$\mathbf{F}_{52} = \mathbf{I}_E \times \mathbf{F} \times \begin{bmatrix} \mathbf{I}_{L \times L} \\ \mathbf{0} \end{bmatrix} = \mathbf{I}_E \times \mathbf{F}_{64}. \tag{3}$$

In (2), $\mathbf{I}_{L \times L}$ is the $(L \times L)$ identity matrix and $\mathbf{I}_T$ is the $(N \times L)$ truncation matrix of extracting first the $L$ column vectors from the full FFT matrix $\mathbf{F}$. In (3), $\mathbf{I}_E$ is the $(52 \times N)$ matrix of extracting 52 row vectors corresponding to the positions of the useful subcarriers in the given $\mathbf{F}_{64}$, defined as

$$\mathbf{I}_E = \begin{bmatrix} \mathbf{0}_{26 \times 5} & \mathbf{I}_{26 \times 26} & \mathbf{0}_{26 \times 1} & \mathbf{0}_{26 \times 26} & \mathbf{0}_{26 \times 6} \\ \mathbf{0}_{26 \times 5} & \mathbf{0}_{26 \times 26} & \mathbf{0}_{26 \times 1} & \mathbf{I}_{26 \times 26} & \mathbf{0}_{26 \times 6} \end{bmatrix}. \tag{4}$$

Moreover, the nulling matrix can be expressed, from $\mathbf{I}_T$ in (2), as

$$\mathbf{I}_N = \mathbf{I}_T \times \mathbf{I}_T^\mathsf{H} = \begin{bmatrix} \mathbf{I}_{L \times L} & \mathbf{0} \\ \mathbf{0} & \mathbf{0} \end{bmatrix} \tag{5}$$

where $\mathbf{I}_N \times \mathbf{A}$ gives all elements, except the first $L$ in the $(N \times 1)$ column vector $\mathbf{A}$, which are nulled.

Note that, in (2), $L$ denotes the CIR length (i.e., the maximum access delay time normalized by the discrete sampling time [12]). When $\hat{L}$ is the estimated maximum access delay time, $\hat{L} = L$ denotes the perfectly estimated CIR length.

### 2.2. OFDM Signal Presentation

By removing the GI and applying the FFT to the received signal, the frequency-domain received signal $(N \times 1)$ vector $\mathbf{Y}_i$ of the $i$th OFDM symbol can be expressed, for the quasi-static fading channels, as

$$
\begin{aligned}
\mathbf{Y}_i &= \mathbf{X}_i \mathbf{H}_i + \mathbf{W}_i \\
&= \mathbf{X}_i \mathbf{F}_{64} \mathbf{h}_i + \mathbf{W}_i
\end{aligned}
\tag{6}
$$

where $i \in \{1, 2, \cdots, M\}$, $Y_i(k)|_{k \in \mathbb{S}_u} = X_i(k,k)H_i(k) + W_i(k)$, $Y_i(k)|_{k \in \mathbb{S}_v} = X_i(k,k)|_{k \in \mathbb{S}_v} = W_i(k)|_{k \in \mathbb{S}_v} = 0$, and $M$ is the number of OFDM symbols in data field (i.e., packet). Then, $\mathbf{H}_i$ and $\mathbf{W}_i$ denote the CFR $(N \times 1)$ vector and the additive white Gaussian noise (AWGN) $(N \times 1)$ vector in which the element $W_i(k)|_{k \in \mathbb{S}_u}$ has a mean of zero and a variance of $\sigma^2$. In addition, $\mathbf{X}_i$ denotes the $(N \times N)$ diagonal matrix in which the diagonal element $X_i(k,k)|_{k \in \mathbb{S}_u}$ is the modulated data symbol for $k \in \mathbb{S}_d$ and the pilot symbol for $k \in \mathbb{S}_p$ where $E[X_i(k,k)] = 0$, $E\left[|X_i(k,k)|^2\right] = 1$ and $E[\cdot]$ denotes the expectation.

In (6), $\mathbf{h}_i$ is the CIR $(L \times 1)$ vector. Then, $\mathbf{H}_i$ stands for the CFR $(N \times 1)$ vector of

$$
\mathbf{H}_i = \mathbf{F}_{64} \times \mathbf{h}_i = \mathbf{F} \times \mathbf{I}_T \times \mathbf{h}_i.
\tag{7}
$$

### 3. Proposed NC-CE Schemes

Note that we propose the NC-CE scheme utilizing the estimated CIR length obtained by the algorithms in [6,7]. This denotes that $L \leftarrow `\hat{L}$ according to [6]' and $L \leftarrow `\hat{L}$ according to [7]'.

### 3.1. Initial Step (Long Preamble CE)

For the convenience of description for the initial channel estimation, let us represent the received signal for two long preambles (i.e., $\mathbf{Y}_i|_{i=-2}^{-1}$ of (6)), from $\mathbf{I}_E$ of (4), as follows [1,10]:

$$
\begin{aligned}
\mathbf{y}_0^{\mathrm{LP}_1} &= \mathbf{I}_E \times \mathbf{Y}_{-2} = \mathrm{diag}\left(\mathbf{x}^{\mathrm{LT}}\right)\mathbf{F}_{52}\mathbf{h}_0 + \mathbf{w}_0^{\mathrm{LP}_1} \\
\mathbf{y}_0^{\mathrm{LP}_2} &= \mathbf{I}_E \times \mathbf{Y}_{-1} = \mathrm{diag}\left(\mathbf{x}^{\mathrm{LT}}\right)\mathbf{F}_{52}\mathbf{h}_0 + \mathbf{w}_0^{\mathrm{LP}_2} \\
\mathbf{w}_0^{\mathrm{LP}_1} &= \mathbf{I}_E \times \mathbf{W}_{-2} \\
\mathbf{w}_0^{\mathrm{LP}_2} &= \mathbf{I}_E \times \mathbf{W}_{-1}
\end{aligned}
\tag{8}
$$

In (8), $\mathbf{x}^{\mathrm{LT}}$ is the long training BPSK symbol $(52 \times 1)$ vector known to the receiver [1]. By the least square (LS) method, the initial CFR $(52 \times 1)$ vector in the useful subcarriers can be obtained as

$$
\hat{\mathbf{H}}_0 = \frac{1}{2}\left(\mathbf{y}_0^{\mathrm{LP}_1} + \mathbf{y}_0^{\mathrm{LP}_2}\right)./\mathbf{x}^{\mathrm{LT}} = \hat{\mathbf{x}}_0^{\mathrm{H}} \mathbf{y}_0^{\mathrm{LP}}
\tag{9}
$$

where $\hat{\mathbf{x}}_0 = \mathrm{diag}\left(\mathbf{x}^{\mathrm{LT}}\right)$ and $\mathbf{y}_0^{\mathrm{LP}} = \frac{1}{2}\left(\mathbf{y}_0^{\mathrm{LP}_1} + \mathbf{y}_0^{\mathrm{LP}_2}\right)$.

As shown in [3,5], we can estimate the CIR vector by invoking the time-domain least square (TDLS) estimation strategy from (9) as

$$
\hat{\mathbf{h}}_0 = \left(\mathbf{F}_{52}^{\mathrm{H}} \hat{\mathbf{x}}_0^{\mathrm{H}} \hat{\mathbf{x}}_0 \mathbf{F}_{52}\right)^{-1} \mathbf{F}_{52}^{\mathrm{H}} \hat{\mathbf{H}}_0
\tag{10}
$$

where $\hat{\mathbf{x}}_0^H \hat{\mathbf{x}}_0 = \mathbf{I}_{52 \times 52}$. Then, the CFR can be obtained by transforming $\hat{\mathbf{h}}_0$ into the frequency domain as

$$\hat{\mathbf{H}}_0^{\text{FSC}} = \mathbf{F}_{64} \hat{\mathbf{h}}_0 = \mathbf{F}_{64} \left( \mathbf{F}_{52}^H \mathbf{F}_{52} \right)^{-1} \mathbf{F}_{52}^H \hat{\mathbf{H}}_0 = \mathbf{Q}_0 \hat{\mathbf{H}}_0. \tag{11}$$

As mentioned in [5], $\mathbf{Q}_0$ indicates a linear filter matrix to estimate the CFR $(N \times 1)$ vector of $\hat{\mathbf{H}}_0^{\text{FSC}}$, corresponding to the virtual subcarriers as well as the useful subcarriers, from $\hat{\mathbf{H}}_0$ of (9), which is expressed as

$$\mathbf{Q}_0 = \mathbf{F}_{64} \left( \mathbf{F}_{52}^H \mathbf{F}_{52} \right)^{-1} \mathbf{F}_{52}^H \tag{12}$$

where $\mathbf{Q}_0 \neq \mathbf{F}_{64} \mathbf{F}_{52}^H$ and $\mathbf{F}_{52}^H \mathbf{F}_{52} \neq \mathbf{I}$. Note that $\hat{\mathbf{H}}_0$ in (9) is the $(52 \times 1)$ column vector and $\hat{\mathbf{H}}_0^{\text{FSC}}$ in (11) is the $(N \times 1)$ column vector.

### 3.2. Modified CDP Step

In [10], the author showed the CDP-based channel estimation scheme with the reliability test. The authors in [4,5] presented the enhanced channel estimation methods based on CDP as a structure with the reliability test and the frequency-domain interpolation. Notice that the proposed CE method adopts the CDP scheme without both the reliability test and the frequency-domain interpolation.

#### 3.2.1. Constructing Data Pilots by LS Method

The received signal of the $i$th data OFDM symbol $\mathbf{Y}_i$ in (6) is equalized using the $(i-1)$th estimated channel value $\hat{\mathbf{H}}_{i-1}^{\text{FSC}}$, and then, by demapping, the constructed data pilot can be expressed as

$$\hat{\mathbf{X}}_i = \text{diag}\left( \mathbf{p}_d D\left( \mathbf{Y}_i. / \hat{\mathbf{H}}_{i-1}^{\text{FSC}} \right) + \mathbf{p}_p \mathbf{X}_i^P \right) + \mathbf{p}_v \tag{13}$$

where $\mathbf{X}_i^P$ denotes the $(N \times 1)$ pilot symbol vector and $D(\cdot)$ is a function that maps the equalized signal with regard to the corresponding modulation order [10]. Note, in (13), that $\hat{X}_i(k,k)\big|_{k \in \mathbb{S}_v} = 1$, $\hat{X}_i(k,k)\big|_{k \in \mathbb{S}_d} = D\left( Y_i(k) / \hat{H}_{i-1}^{\text{FSC}}(k) \right)$, and $\hat{X}_i(k,k)\big|_{k \in \mathbb{S}_p} = X_i^P(k)$, which is a predefined frequency-domain pilot symbol of the $k$th subcarrier in the $i$th OFDM symbol.

The instantaneous channel coefficient $\hat{\mathbf{H}}_i$ can be expressed by equalizing $\mathbf{Y}_i$ of (6) with $\hat{\mathbf{X}}_i$ of (13) as

$$\hat{\mathbf{H}}_i = \mathbf{Y}_i. / \hat{\mathbf{X}}_i \tag{14}$$

with $Y_i(k)\big|_{k \in \mathbb{S}_v} = 0$, $\hat{H}_i(k)\big|_{k \in \mathbb{S}_v} = 0$ and $\hat{H}_i(k)\big|_{k \in \mathbb{S}_u} = Y_i(k) / \hat{X}_i(k,k)$.

#### 3.2.2. Virtual Subcarrier Filling Step

Then, we make the frequency-domain $N$-point CFRs not only perform the effective noise cancellation but also implement the IFFT/FFT operations. Unlike conventional CE schemes (i.e., the 52-point type [11]), we leverage the virtual CFRs estimated in the previous OFDM symbol time (i.e., $i-1$) for $k \in \mathbb{S}_v$ to improve the noise cancellation. We can express the estimated CFRs in the current $i$th OFDM symbol as

$$\bar{\mathbf{H}}_i = \mathbf{p}_v \hat{\mathbf{H}}_{i-1}^{\text{FSC}} + \mathbf{p}_u \hat{\mathbf{H}}_i. \tag{15}$$

Note that the CE method in [5] with $\hat{L} = (L_{\text{cp}} - 1)$ has an enhanced TRFI step similar to [11], but the proposed CE methods with $\hat{L}$ by [6] or by [7] do not require an additional interpolation step, such as the TRFI [11].

### 3.3. Noise-Canceling Step

In this step, we attenuate the noise components in (6) and the noisy CFRs in (15) as

$$\hat{\mathbf{H}}_i^{\text{FSC}} = \mathbf{F}_{64}\left(\mathbf{F}_{64}^{\mathsf{H}}\mathbf{F}_{64}\right)^{-1}\mathbf{F}_{64}^{\mathsf{H}}\bar{\mathbf{H}}_i = \mathbf{Q}_D\bar{\mathbf{H}}_i \tag{16}$$

where $\mathbf{Q}_D = \mathbf{F}_{64}\mathbf{F}_{64}^{\mathsf{H}}$ and $\mathbf{F}_{64}^{\mathsf{H}}\mathbf{F}_{64} = \mathbf{I}_{L\times L}$. Note that $\hat{\mathbf{H}}_0$ of (9) is the $(52 \times 1)$ vector and $\hat{\mathbf{H}}_i\big|_{i=1}^{M}$ of (14) is the $(64 \times 1)$ vector. Furthermore, from $\mathbf{F}_{64}$ of (2) and $\mathbf{I}_N$ of (5), the operation of (16) can be equivalently expressed as

$$\begin{aligned}
\hat{\mathbf{H}}_i^{\text{FSC}} = \mathbf{F}_{64}\mathbf{F}_{64}^{\mathsf{H}}\bar{\mathbf{H}}_i &= (\mathbf{F} \times \mathbf{I}_T)(\mathbf{F} \times \mathbf{I}_T)^{\mathsf{H}}\bar{\mathbf{H}}_i \\
&= \mathbf{F} \times \mathbf{I}_T(\mathbf{I}_T)^{\mathsf{H}}\mathbf{F}^{\mathsf{H}}\bar{\mathbf{H}}_i \\
&= \mathbf{F} \times \mathbf{I}_N \times \mathbf{F}^{\mathsf{H}} \times \bar{\mathbf{H}}_i
\end{aligned} \tag{17}$$

and this means that $\hat{\mathbf{H}}_i^{\text{FSC}}$ can be obtained from $\bar{\mathbf{H}}_i$ of (15) through the IFFT, truncation (nulling), and FFT operations as follows:

- $\hat{\mathbf{h}}_i = \mathbf{F}^{\mathsf{H}} \times \bar{\mathbf{H}}_i$ : By the IFFT, the CFR can be transformed into the time-domain CIR.
- $\mathbf{I}_N \times \hat{\mathbf{h}}_i$ : Noise cancellation in the time domain is performed by nulling components that exceed the $L$ row in the CIR of $\hat{\mathbf{h}}_i$.
- $\mathbf{F} \times (\mathbf{I}_N \times \hat{\mathbf{h}}_i)$ : By the FFT, the noise-canceled CIR can be transformed into the CFR.

For the next $(i+1)$th data field, we can go to the step of (13). The process is repeated until we arrive at the $M$th OFDM symbol.

Let us compare (16) to (11). $\hat{\mathbf{H}}_0$ in (11) does not have a virtual subcarrier component, and $\hat{\mathbf{H}}_0^{\text{FSC}}$ has the estimated virtual subcarrier component. However, $\bar{\mathbf{H}}_i$ in (16) has the virtual subcarrier component $\mathbf{p}_v\hat{\mathbf{H}}_{i-1}^{\text{FSC}}$, determined in the previous step, as shown in (15), and then, the virtual subcarrier component of $\hat{\mathbf{H}}_i^{\text{FSC}}$ is newly obtained by removing noise effects. Furthermore, it can be seen that filling the virtual subcarrier component enables IFFT/FFT-based channel estimation, as shown in (17).

### 3.4. Comparison with TDLS Scheme [3]

From (11) in [5], $\mathbf{y}_i = \mathbf{I}_E\mathbf{Y}_i$ and $\hat{\mathbf{x}}_i = \text{diag}\left(\mathbf{I}_E \times \text{diag}(\hat{\mathbf{X}}_i)\right)$, we can estimate the CFR vector by invoking the TDLS estimation strategy as

$$\hat{\mathbf{H}}_i^{\text{TDLS}} = \mathbf{F}_{52}\left(\mathbf{F}_{52}^{\mathsf{H}}\hat{\mathbf{x}}_i^{\mathsf{H}}\hat{\mathbf{x}}_i\mathbf{F}_{52}\right)^{-1}\mathbf{F}_{52}^{\mathsf{H}}\hat{\mathbf{x}}_i^{\mathsf{H}}\mathbf{y}_i. \tag{18}$$

Notice that when $\hat{\mathbf{x}}_i^{\mathsf{H}}\hat{\mathbf{x}}_i = \mathbf{I}$ (e.g., MPSK) in (18), we can rewrite (18) as

$$\hat{\mathbf{H}}_i^{\text{TDLS}}\Big|_{\text{MPSK}} = \mathbf{F}_{52}\left(\mathbf{F}_{52}^{\mathsf{H}}\mathbf{F}_{52}\right)^{-1}\mathbf{F}_{52}^{\mathsf{H}}\hat{\mathbf{x}}_i^{\mathsf{H}}\mathbf{y}_i = \mathbf{Q}_0\hat{\mathbf{x}}_i^{\mathsf{H}}\mathbf{y}_i \tag{19}$$

where $\mathbf{Q}_0$ of (12) can be used for the data field. On the other hand, for the case of $\hat{\mathbf{x}}_i^{\mathsf{H}}\hat{\mathbf{x}}_i \neq \mathbf{I}$ (e.g., 16QAM and 64QAM) in (18), it has high complexity as the inverse matrix operation must be performed for each OFDM symbol. Moreover, we cannot use the IFFT/FFT operation in both (18) and (19).

### 3.5. Computational Complexity

Related to the computational complexity, the proposed NC-CE schemes have the following key features:

- $\mathbf{Q}_0$ : It is observed that $\mathbf{Q}_0$ is used only once within the data filed (i.e., per packet) and can be pre-defined offline.
- The IFFT/nulling/FFT operation: It can be implemented with very low complexity for a large $N$. The matrix inversion per the OFDM symbol is not required for 16QAM and 64QAM.

- An interpolation scheme (e.g., TRFI) is not required.

Table 1 compares the complexity of the proposed NC-CE schemes and the existing CE methods from [3–5].

**Table 1.** Legend description for simulation results and complexity comparison.

| Comments | Legend | $\hat{L}$ | Complexity (MI [1], TRFI) | IFFT/Nulling/FFT |
|---|---|---|---|---|
| | `Ideal` | $L$ | | |
| Bounds | `TDLS + Ideal`$(\hat{L} = L)$ | $L$ | High (O,X) | X |
| | `Ref.[5] + Ideal`$(\hat{L} = L)$ | $L$ | Medium (X,O) | O [2] |
| [3] | `TDLS +` $\hat{L} = (L_{cp} - 1)$ | $L_{cp} - 1$ | High (O,X) | X |
| [5] | `Ref.[5] +` $\hat{L} = (L_{cp} - 1)$ | $L_{cp} - 1$ | Medium (X,O) | O [2] |
| [4] | `Ref.[4] +` $\hat{L} = (L_{cp} - 1)$ | $L_{cp} - 1$ | Medium (X,O) | X |
| Proposed | `Prop. +` $\hat{L}$ by [6] | by [6] | Low (X,X) | O |
| Methods | `Prop. +` $\hat{L}$ by [7] | by [7] | Low (X,X) | O |
| [3,6] | `TDLS +` $\hat{L}$ by [6] | by [6] | High (O,X) | X |
| [3,7] | `TDLS +` $\hat{L}$ by [7] | by [7] | High (O,X) | X |

[1] Matrix inversion (MI) per OFDM symbol is required for 16QAM and 64QAM. [2] Although not mentioned in [5], implementation by the IFFT/nulling/FFT operations can be confirmed through the analytical approach of this paper.

## 4. Simulation Results

In this section, we present simulation results to verify the error rates and the mean square error (MSE) performance of the proposed scheme based on the IEEE 802.11p standard with $N = 64$, $L_{cp} = 16$, and $T_s = 0.1$ μs [1,12]. In Table 3 in [12], we can see the key parameters of IEEE 802.11p used in simulations. The transmitter and the receiver basically adopt the convolutional encoder and the Viterbi decoder with constraint length of seven for both [1,12]. We assume that one packet consists of $M$ (=100) OFDM symbols, QPSK, 16QAM, and 64QAM with a coding rate of one-half. For all cases, over $5 \times 10^5$ packet transmissions with SNR $= E\left[|H_i(k)|^2\right]/\sigma^2$ were averaged. The error rate comparison is carried out in terms of the packet error rate (PER) and the coded bit error rate (BER). Among the five scenarios of 'CohdaWireless V2V channel model' in [17], we consider the 'Street Crossing NLOS with 126 km/h' and 'Highway LOS with 252km/h' channel environments. The other parameters, such as the delay time, relative power, and Doppler spectrum for each channel tap, are listed in [17].

As shown in Table 1, three performance bounds, 'Ideal', 'TDLS + Ideal $(\hat{L} = L)$', and 'Ref. [5] + Ideal $(\hat{L} = L)$', are presented for performance comparison. 'Ideal' and 'TDLS + Ideal $(\hat{L} = L)$' denote that the channel coefficient obtained by the FFT on the actual time-varying channel value at the middle position of each OFDM symbol (i.e., $\mathbf{H}_i$ in (7)) and the TDLS CE method in [3] having perfectly estimated $L$. Also, 'Ref. [5] + Ideal $(\hat{L} = L)$' denotes the CE method in [5], having both the TRFI step of [11] and a perfectly estimated $L$. In addition, we consider the three existing CE methods of [3–5], which are 'TDLS + $\hat{L} = (L_{cp} - 1)$', 'Ref. [5] + $\hat{L} = (L_{cp} - 1)$', and 'Ref. [4] + $\hat{L} = (L_{cp} - 1)$'. 'TDLS + $\hat{L} = (L_{cp} - 1)$' denotes the TDLS CE method in [3]. 'Ref. [5] + $\hat{L} = (L_{cp} - 1)$' and 'Ref. [4] + $\hat{L} = (L_{cp} - 1)$' are the CE method in [5], which have the TRFI step of [11] and the CE method in [4], respectively. Furthermore, 'Prop. + $\hat{L}$ by [6]' and 'Prop. + $\hat{L}$ by [7]' represent the proposed CE methods of (17) with the estimated channel length $\hat{L}$ to be obtained by the algorithms in [6,7], respectively. Also, 'TDLS + $\hat{L}$ by [6]' and 'TDLS+$\hat{L}$ by [7]' represent the TDLS CE methods with the estimated channel length $\hat{L}$ by [6,7], respectively. Table 1 illustrates the relationship among the variable $\hat{L}$, the complexity, and the implementation of the IFFT/nulling/FFT for all of the considered CE methods in the simulations.

### 4.1. Simulation Results for MSE

From here, let us define the MSE of the $k$th CFR and the average MSE (AMSE) of the CFRs as

$$\text{MSE}_k = E\left[\left|\hat{H}_i(k) - H_i(k)\right|^2\right]$$

$$\text{AMSE} = \frac{1}{N_u} \sum_{k \in \mathbb{S}_u} E\left[\left|\hat{H}_i(k) - H_i(k)\right|^2\right], \quad (20)$$

where $\hat{H}_i(k)$ is the estimated $k$th CFR for the given scheme and $H_i(k)$ is the ideal $k$th CFR obtained by the FFT on the actual time-varying channel coefficient at the middle position of each OFDM symbol (i.e., $\mathbf{H}_i$ in (7)).

Figures 1–3 show the $k$th CFR MSE comparison with respect to CE schemes under 'Street Crossing NLOS with 126 km/h' for QPSK, 16QAM, and 64QAM, respectively. Figures 4–6 show the $k$th CFR MSE comparison with respect to CE methods under 'Highway LOS with 252 km/h' for QPSK, 16QAM, and 64QAM, respectively. Figures 7 and 8 show the AMSE comparison with respect to CE schemes under 'Street Crossing NLOS with 126 km/h' and 'Highway LOS with 252 km/h', respectively. From Figures 1–8, note that the three TDLS schemes (i.e., TDLS + Ideal ($\hat{L} = L$), TDLS + $\hat{L}$ by [6], and TDLS + $\hat{L}$ by [7]) do not estimate the CFRs for virtual subcarriers.

From Figures 1–8, the proposed schemes (red lines) are observed achieving the MSE bounds (cyan lines) of 'TDLS + $\hat{L}$ by [6]' and 'TDLS + $\hat{L}$ by [7]'. Especially near the edge of both virtual subcarriers, the MSEs of the proposed schemes not only approach the achievable bound of 'Ref. [5] + Ideal ($\hat{L} = L$)' but also outperform the 'TDLS + Ideal ($\hat{L} = L$)' (e.g., Figures 1b,c and 2b,c). Therefore, it can be seen that at some SNR points the proposed scheme can provide slightly lower error rates than 'TDLS + $\hat{L}$ by [6]' and 'TDLS + $\hat{L}$ by [7]'. Consequently, in general, the three existing CE schemes (black lines) with $\hat{L} = (L_{cp} - 1)$ give a larger MSE near the edge of both virtual subcarriers. However, it can be seen from the proposed schemes (red lines) that all frequency regions exhibit low MSEs, regardless of the SNR.

Notice that the nulling (i.e., noise canceling) by $\mathbf{I}_N$ in (17) can be comprehended as the circular convolution of the complex sinc function on the frequency domain, which is shown in Figures 9 and 10 for $\hat{L} = 7$ and $\hat{L} = L_{cp} - 1 = 15$, respectively. In instances where the reliability of certain CFR information is compromised in proximity to the boundary of data subcarriers, the enhancement of CE performance can be achieved through frequency-domain filtering, in which filter coefficients are properly estimated as $\hat{L} = L$. Notably, the proposed method emphasizes the preservation of circular symmetry in the estimated CFR values, a characteristic conducive to improved CE accuracy. This approach contrasts with existing schemes employing $\hat{L} = (L_{cp} - 1)$, wherein the filtering is executed utilizing filter coefficients characterized by rapidly rotating phase components, as shown in Figure 10c. The consequence of such rotations is the generation of an elevated MSE in the vicinity of virtual subcarriers (e.g., Figures 1b,c and 4b,c).

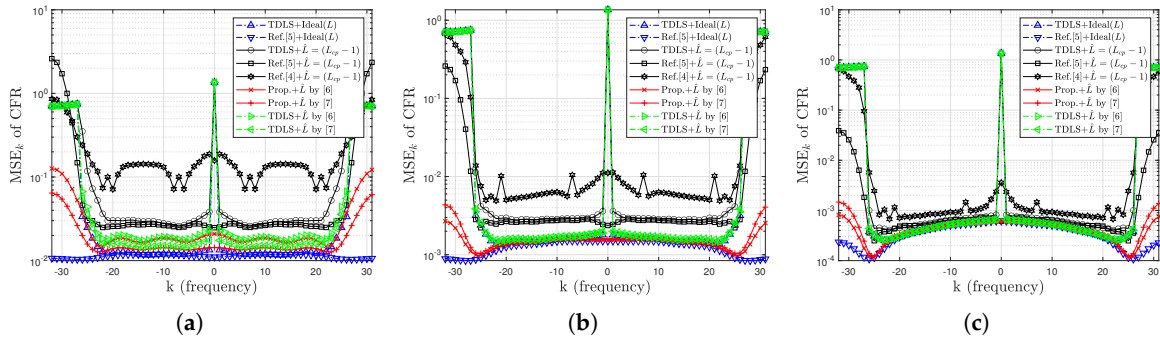

**Figure 1.** MSE of CFR comparison at Street Crossing NLOS (126 km/h, $L = 7$, QPSK, CR = 1/2, $M = 100$): (**a**) SNR = 10 dB, (**b**) SNR = 20 dB, and (**c**) SNR = 30 dB [4–7].

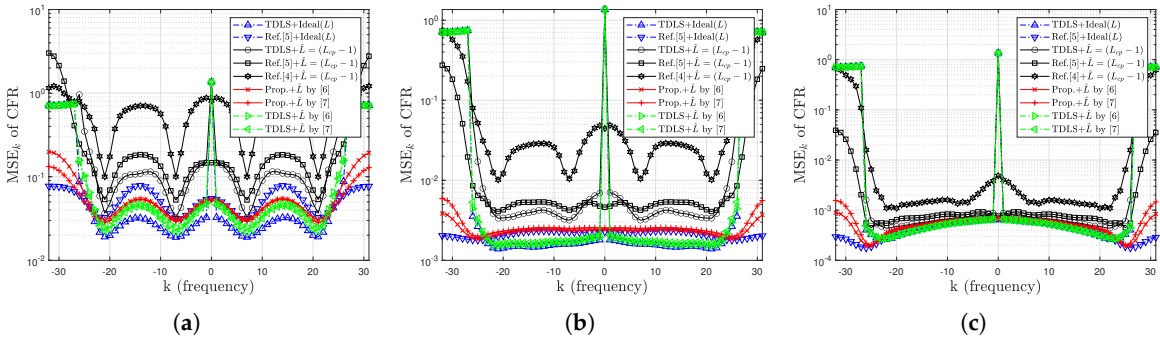

**Figure 2.** MSE of CFR comparison at Street Crossing NLOS (126 km/h, $L = 7$, 16QAM, CR = 1/2, $M = 100$): (**a**) SNR = 10 dB, (**b**) SNR = 20 dB, and (**c**) SNR = 30 dB [4–7].

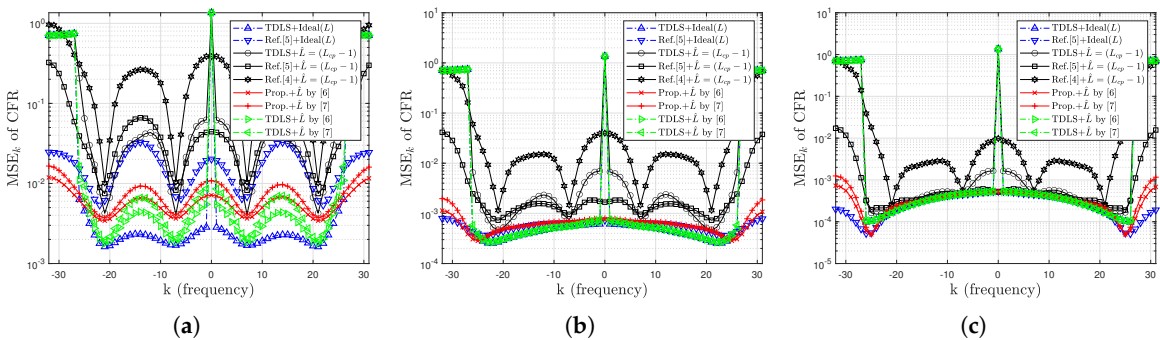

**Figure 3.** MSE of CFR comparison at Street Crossing NLOS (126 km/h, $L = 7$, 64QAM, CR = 1/2, $M = 100$): (**a**) SNR = 10 dB, (**b**) SNR = 20 dB, and (**c**) SNR = 30 dB [4–7].

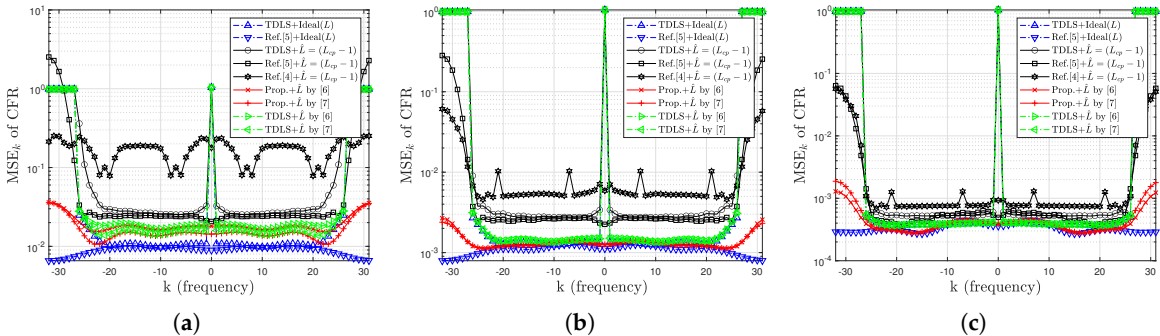

**Figure 4.** MSE of CFR comparison at Highway LOS (252 km/h, $L = 6$, QPSK, CR = 1/2, $M = 100$): (**a**) SNR = 10 dB, (**b**) SNR = 20 dB, and (**c**) SNR = 30 dB [4–7].

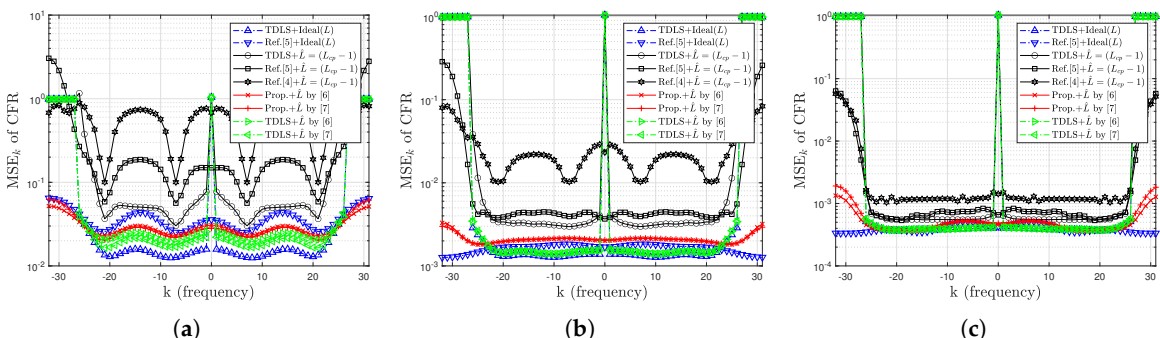

**Figure 5.** MSE of CFR comparison at Highway LOS (252 km/h, $L = 6$, 16QAM, CR = 1/2, $M = 100$): (**a**) SNR = 10 dB, (**b**) SNR = 20 dB, and (**c**) SNR = 30 dB [4–7].

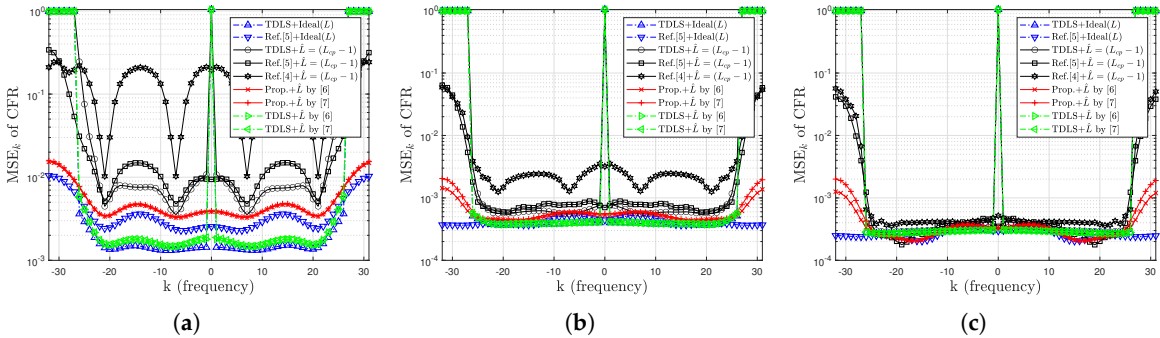

**Figure 6.** MSE of CFR comparison at Highway LOS (252 km/h, $L = 6$, 64QAM, CR = 1/2, $M = 100$): (**a**) SNR = 10 dB, (**b**) SNR = 20 dB, and (**c**) SNR = 30 dB [4–7].

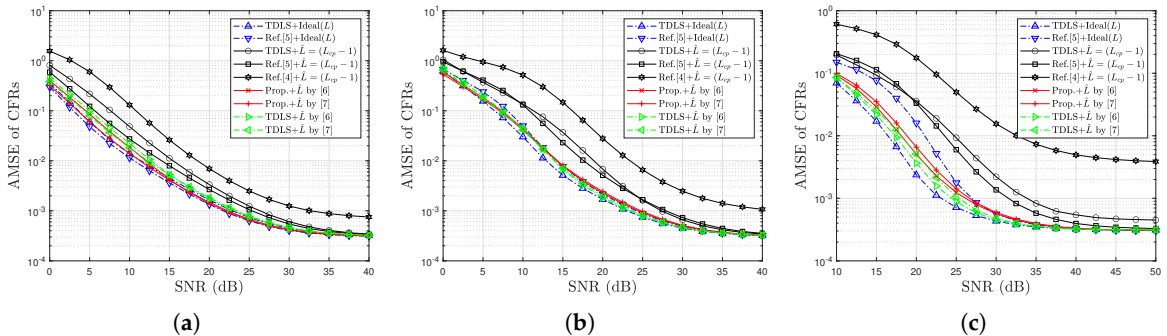

**Figure 7.** AMSE of CFR comparison at Street Crossing NLOS (126 km/h, $L = 7$, QPSK, 16QAM, 64QAM, CR = 1/2, $M = 100$): (**a**) QPSK, (**b**) 16QAM, and (**c**) 64QAM [4–7].

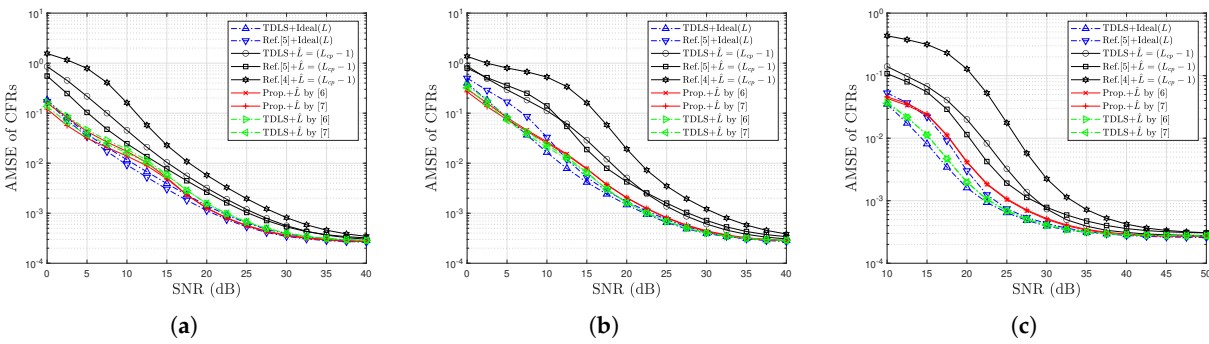

**Figure 8.** AMSE of CFR comparison at Highway LOS (252 km/h, $L = 6$, QPSK, 16QAM, 64QAM, CR = 1/2, $M = 100$): (**a**) QPSK, (**b**) 16QAM, and (**c**) 64QAM [4–7].

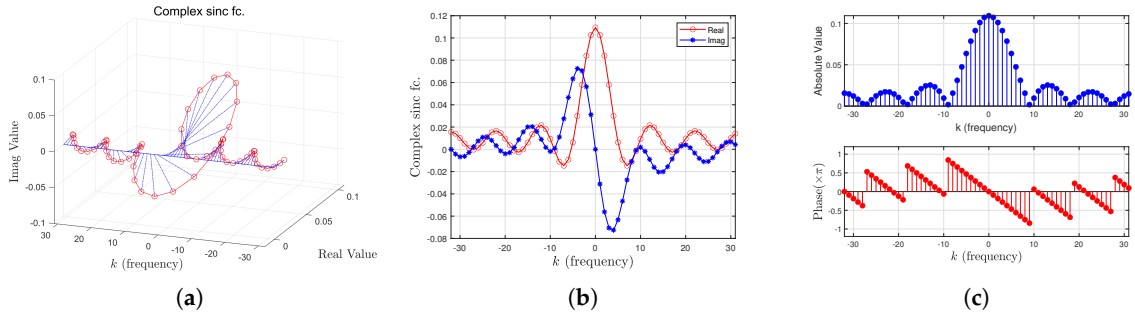

**Figure 9.** Complex sinc function ($\hat{L} = 7$): (**a**) complex presentation, (**b**) real ($\Re$) and imag ($\Im$), and (**c**) magnitude and phase.

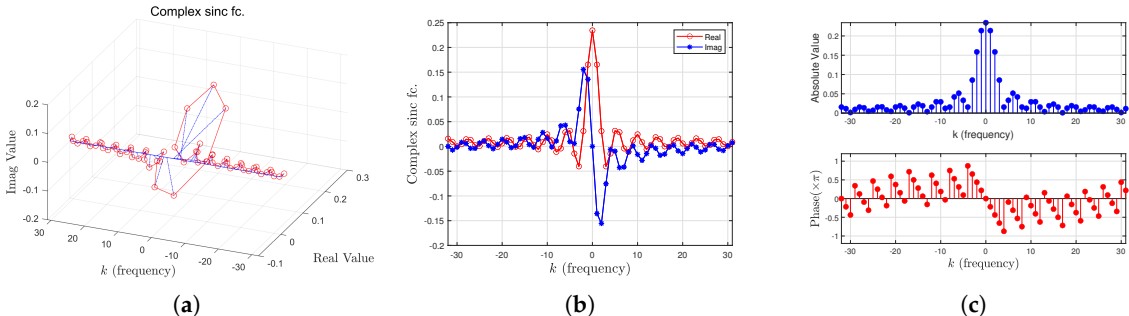

**Figure 10.** Complex sinc function ($\hat{L} = L_{\text{cp}} - 1 = 15$): (**a**) complex presentation, (**b**) real ($\Re$) and imag ($\Im$), and (**c**) magnitude and phase.

### 4.2. Simulation Results for Error Rates

Figures 11–13 show the error rate performance comparison with respect to CE schemes under 'Street Crossing NLOS with 126 km/h' for QPSK, 16QAM, and 64QAM, respectively. Figures 14–16 show the error rate performance comparison with respect to CE methods under 'Highway LOS with 252km/h' for QPSK, 16QAM, and 64QAM, respectively.

In Figures 11 and 14 (i.e., QPSK case), we observe that the proposed schemes (red lines) can outperform the three existing schemes that have $\hat{L} = (L_{\text{cp}} - 1)$ (black lines). Moreover, the proposed schemes give lower PERs than the achievable bounds (cyan lines) of the TDLS schemes with high complexity at some SNR points. Better MSE performances can be interpreted near the edge of both virtual subcarriers, as observed in Figures 1 and 4.

For the 16QAM case (i.e., Figures 12 and 15), we can see that the achievable bounds (cyan lines) of the TDLS schemes approach the ideal case. This means that the *L* estimation performance in [6,7] converges to the ideal case (i.e., $\hat{L} = L$). Then, the proposed schemes (red lines) converge to the achievable bounds (cyan lines) of the TDLS schemes. The same trend can also be seen in Figures 7b and 8b. Although the error floor of the proposed method can be observed in 64QAM under the high SNR region of the NLOS environment (see Figure 13a), it can be said that overall it shows the same trend as that observed in 16QAM.

From the six figures, we can generally say that the performance of the proposed schemes (red lines) approaches three ideal cases (i.e., 'Ideal', 'TDLS + Ideal ($\hat{L} = L$)', and 'Ref. [5] + Ideal ($\hat{L} = L$)'. In addition, the proposed schemes (red lines) can outperform the three existing schemes (black lines) that have $\hat{L} = (L_{\text{cp}} - 1)$ over all practical SNR ranges. Furthermore, the proposed schemes (red lines) are observed to achieve the performance bounds (cyan lines) of 'TDLS + $\hat{L}$ by [6]' and 'TDLS + $\hat{L}$ by [7]' (which requires high complexity associated with a matrix inversion in 16QAM and 64QAM).

Table 2 shows the SNR gains of 'Prop.+$\hat{L}$ by [7]' over 'Ref. [5] + $\hat{L} = (L_{\text{cp}} - 1)$'. It is confirmed that the proposed NC-CE schemes without the TRFI step can obtain SNR gains over the existing method [5] that has the TRFI step. Also, it can be seen that the performance gain increases in 16QAM (or 64QAM) rather than in QPSK.

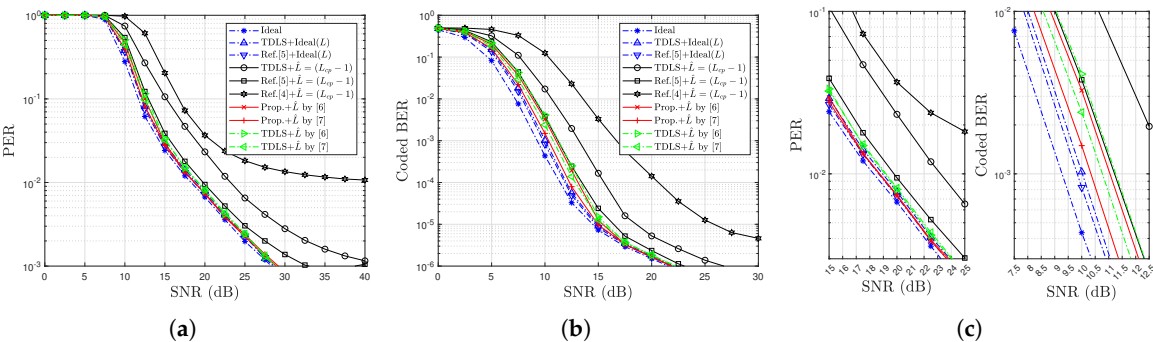

**Figure 11.** Error performance comparison at Street Crossing NLOS (126 km/h, $L = 7$, QPSK, CR = 1/2, $M = 100$): (**a**) PER vs. SNR (dB), (**b**) BER vs. SNR (dB), and (**c**) ERs vs. SNR (dB) [4–7].

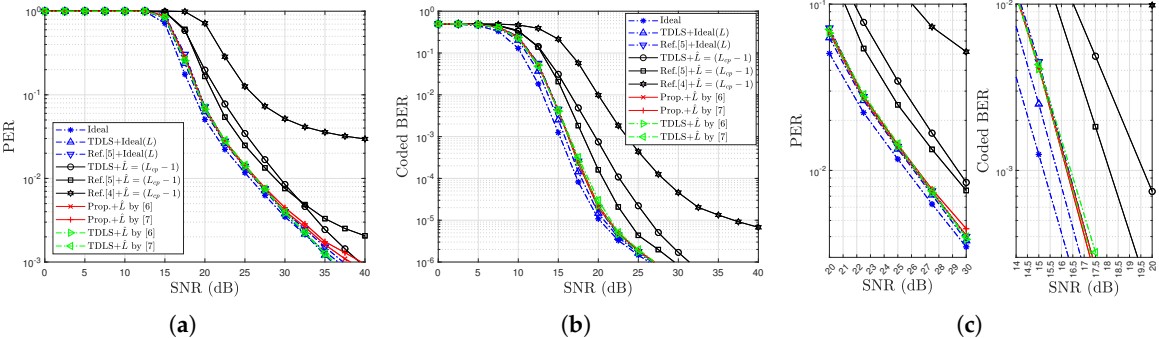

**Figure 12.** Error performance comparison at Street Crossing NLOS (126 km/h, $L = 7$, 16QAM, CR = 1/2, $M = 100$): (**a**) PER vs. SNR (dB), (**b**) BER vs. SNR (dB), and (**c**) ERs vs. SNR (dB) [4–7].

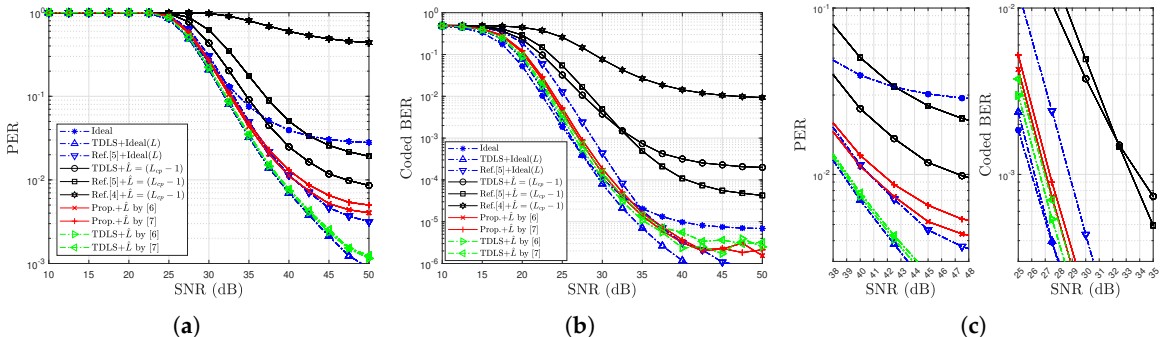

**Figure 13.** Error performance comparison at Street Crossing NLOS (126 km/h, $L = 7$, 64QAM, CR = 1/2, $M = 100$): (**a**) PER vs. SNR (dB), (**b**) BER vs. SNR (dB), and (**c**) ERs vs. SNR (dB) [4–7].

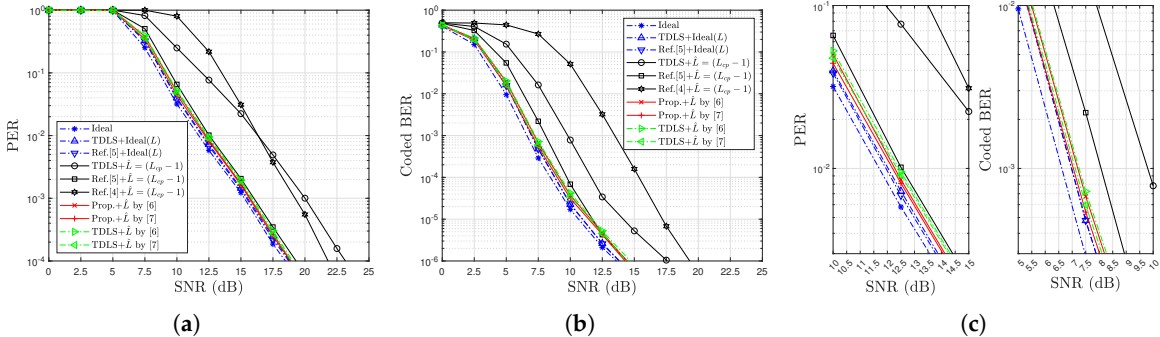

**Figure 14.** Error performance comparison at Highway LOS (252 km/h, $L = 6$, QPSK, CR = 1/2, $M = 100$): (**a**) PER vs. SNR (dB), (**b**) BER vs. SNR (dB), and (**c**) ERs vs. SNR (dB) [4–7].

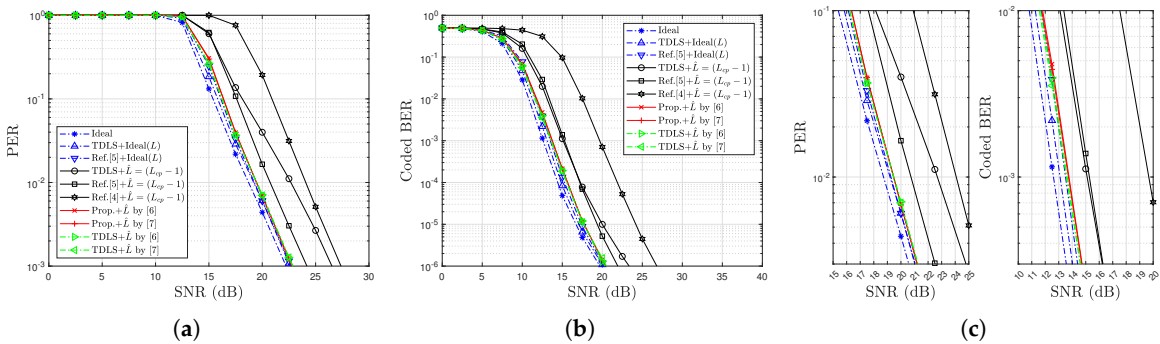

**Figure 15.** Error performance comparison at Highway LOS (252 km/h, $L = 6$, 16QAM, CR = 1/2, $M = 100$): (**a**) PER vs. SNR (dB), (**b**) BER vs. SNR (dB), and (**c**) ERs vs. SNR (dB) [4–7].

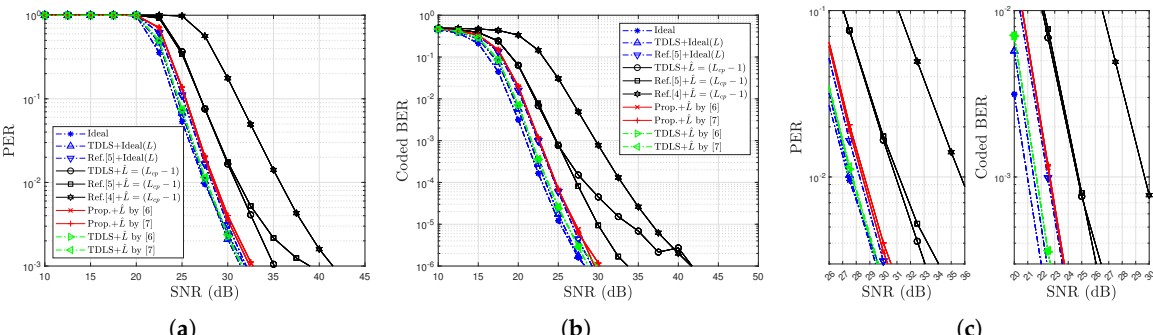

**Figure 16.** Error performance comparison at Highway LOS (252 km/h, $L = 6$, 64QAM, CR = 1/2, $M = 100$): (**a**) PER vs. SNR (dB), (**b**) BER vs. SNR (dB), and (**c**) ERs vs. SNR (dB) [4–7].

**Table 2.** SNR gain (dB) of 'Prop. + $\hat{L}$ by [7]' over 'Ref. [5] + $\hat{L} = (L_{cp} - 1)$'.

| Environment | At PER = $10^{-2}$ | At BER = $10^{-3}$ | From |
|:---:|:---:|:---:|:---:|
| Street Crossing NLOS, QPSK | 1.0 dB | 0.9 dB | Figure 11c |
| Street Crossing NLOS, 16QAM | 2.5 dB | 1.8 dB | Figure 12c |
| Street Crossing NLOS, 64QAM | $\gg$5.5 dB | 6.0 dB | Figure 13c |
| Highway LOS, QPSK | 0.25 dB | 1.0 dB | Figure 14c |
| Highway LOS, 16QAM | 1.3 dB | 1.5 dB | Figure 15c |
| Highway LOS, 64QAM | 2.6 dB | 2.0 dB | Figure 16c |

## 5. Conclusions

In this paper, we have studied novel NC-CE methods that not only outperform the existing designs with $\hat{L} = (L_{cp} - 1)$ from [3–5] but also approach the achievable performance bounds in all practical SNR ranges. To this end, we develop a three-step channel estimation process. In the first step, we adopt the noise-attenuating and virtual subcarrier estimating scheme for long preamble parts. Then, in the second step, we develop the modified CDP structure without any interpolation method, filling the CFRs of the virtual subcarriers obtained in the previous OFDM symbol to cancel the noise components from the instantaneously estimated noisy CFRs. In the final step, we cancel the noise components from the CFRs via the IFFT/nulling/FFT operation, which enables low-complexity implementation. Finally, the simulation results verify the efficiency of our proposed method.

In this paper, it is assumed that the power delay profile of the channel does not change within one packet. Moreover, the proposed CE method relies heavily on the accuracy of the estimated $L$. Therefore, as a future research topic, we suggest the study of the efficient $L$

estimation scheme and then, verification of the performance the proposed CE technique in an environment where the power delay profile of the channel changes within one packet.

**Author Contributions:** Conceptualization, K.K. and H.W.; methodology, H.W.; software, K.K.; validation, K.K. and H.W.; formal analysis, K.K.; investigation, K.K. and H.W.; resources, K.K.; data curation, K.K.; writing—original draft preparation, K.K.; writing—review and editing, H.W.; visualization, K.K.; supervision, H.W.; project administration, K.K. and H.W.; funding acquisition, K.K. and H.W. All authors have read and agreed to the published version of the manuscript.

**Funding:** This work was supported by the National Research Foundation of Korea (NRF) grants funded by the Korean government (MSIT) (NRF-2020R1A2C1005260 and NRF-2021R1A2C1014063).

**Data Availability Statement:** Data are contained within the article.

**Conflicts of Interest:** The authors declare no conflict of interest.

## Abbreviations

The following abbreviations are used in this manuscript:

| | |
|---|---|
| AWGN | additive white Gaussian noise |
| BER | bit error rate |
| CDP | constructed data pilots |
| CFR | channel frequency response |
| CIR | channel impulse response |
| FFT | fast Fourier transform |
| GI | guard interval |
| IFFT | inverse fast Fourier transform |
| ISI | inter-symbol interference |
| MSE | mean square error |
| NC-CE | noise-canceling channel estimation |
| OFDM | orthogonal frequency division multiplexing |
| TRFI | time-domain reliability test and frequency-domain interpolation |
| TDLS | time-domain least square |
| PER | packet error rate |

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
