# Peer review of "Noise-Canceling Channel Estimation Schemes Based on the CIR Length Estimation for IEEE 802.11p/OFDM Systems"

_electronics, doi:10.3390/electronics13061110_

Round 1
Reviewer 1 Report
Comments and Suggestions for Authors
Please see the attached file.

Author Response
Point 1: The formatting and numbering of manuscript are problematic. The sequence numbering is inaccurate, as evident from the lack of a connection between line 88 on the second page and line 89 on the third page.
Response 1: We appreciate your careful advice. In this paper, we present the discrete signal model as the matrix form with respect to the virtual subcarriers, data subcarriers, and pilot subcarriers. In order to this, we define the diagonal matrices pp of (11), pv, and pd so that we can express (13), (14), and (15) by utilizing them.
Point 2: In equation (1) on page 3, are ‘r’ and ‘c’ referring to rows and columns? There is no clear definition or description provided.
Response 2: We appreciate your careful advice. Based on your recommendation, we correct it.
Point 3: In equation (11) on page 4, is the multiplication involving Q valid? The dimension of h_0 is related to the number fo effective subcarriers, but is it reasonable to directly use the dimension of the entire subcarrier FFT for multiplication? Does it require zero-padding or other operations? The description is not sufficiently detailed.
Response 3:
We double check the related equations which have shown in [3] and [5]. Note that in (10)~(12), we have to use the reduced FFT matrix of F_64 (64 X L) and F_54 (54 X L) which are different with the full FFT matrix F_64 (64 X 64). Also, at the end of Subsection 3.3, we explain the key features related with (11), (15), and (16).
Point 4: The section from 170 to 182 on the page 10 excessively relies on numerous references, resulting in a confusing narrative and making it challenging for readers to follows. Similar issues are present in earlier sections, contributing to difficulty in comprehending the article. Could you adopt a more explicit logically coherent way of expression.
Response 4: We appreciate your valuable advice. Based on your recommendation, we have revised the earlier manuscript to response reviewers’ comments clearly. In table 1, we add ‘Comments’ column order to clearly specify which is the proposed technique and which are the existing ones. Futhermore, we have revised the relevant part of Section 4 in this regard.
Point 5: In some figurers, the legend is causing noticeable overlap with the result curves, as observed in Figure 13(b) on page 12. Please consider reducing the size of the legend.
Response 5:
We appreciate your careful advice. Based on your recommendation, we change it.
Point 6: The manuscript involves extensive simulation work, demonstrating the author’s meticulous efforts in ths sapect. However, the descriptions of the phenomena represented by the simulation graphs appear to be insufficient. It is recommended to enhance the detailed description of the simulation results to provide a clearer expression of the phenomena reflected in the figures.
Response 6:
We appreciate your careful advice. Based on your recommendation, we added an explanation of the simulation results to clarify the claims of this paper.
Point 7: The introduction section lacks sufficient description of the current research work and requires some supplementation. For imstance, incoperation information on noise elimination method employed in other 802.11 protocols could highlight the innovation presented in this manuscript.
Response 1: We appreciate your valuable advice. Based on your recommendation, we have revised the earlier manuscript to response reviewers’ comments clearly. Furthermore, we made our best effort to more clearly describe the proposed methodology in the abstract, introduction section, and conclusiosn section. As a result, we could improve the revised manuscript based on your advice and reviewer’s comments.
Point 8: The references are predominantly dated, and it is recommended to incorporate more recent literature to showcase the recent developments in relevant technologies.
Response 1: We appreciate your valuable advice. Based on your recommendation, we have revised the earlier manuscript to response reviewers’ comments clearly. Furthermore, we made our best effort to more clearly describe the proposed methodology in the abstract, introduction section, and conclusiosn section. As a result, we could improve the revised manuscript based on your advice and reviewer’s comments.

Reviewer 2 Report
Comments and Suggestions for Authors
This paper investigates novel Noise-cancellation Channel Estimation (NC-CE) methods within the context of IEEE 802.11p standard, specifically focusing on scenarios with different channel models and varying speeds. The authors propose a three-step channel estimation process, demonstrating its efficacy through simulation results and comparisons with existing methods. Overall, the manuscript is well-structured, and the authors have made a substantial effort to address challenges in channel estimation, particularly in vehicular communication environments. However, there are both strong and weak points that need consideration.
++ The paper introduces a novel approach to NC-CE, departing from the conventional ˆL = (Lcp − 1) designs. The proposed three-step process involving noise attenuation, virtual subcarrier estimation, and a modified CDP structure is innovative and addresses the shortcomings of existing methods.
++ The authors conducted extensive simulations under various scenarios, considering different modulation schemes and channel models. The inclusion of performance comparisons with existing methods, such as TDLS and Ref.[5], adds scientific rigor to the evaluation. The use of Mean Square Error (MSE) and error rates provides a thorough assessment of the proposed NC-CE methods.
++ The results are well-presented through multiple figures, providing a clear visualization of the performance of the proposed NC-CE methods. The use of error performance metrics and Signal-to-Noise Ratio (SNR) gains facilitates a detailed understanding of the method's effectiveness in different scenarios.
-- Although the authors mention the low-complexity implementation of their method, a more in-depth discussion on the computational complexity and resource requirements would strengthen the scientific validity of the proposed approach. This is crucial for practical considerations, especially in real-time communication systems.
-- The paper lacks a thorough discussion of potential limitations and challenges associated with the proposed NC-CE methods. Addressing these limitations, even briefly, would contribute to a more balanced and nuanced presentation of the research findings.
Author Response
Point 1: Although the authors mention the low-complexity implementation of their method, a more in-depth discussion on the computational complexity and resource requirements would strengthen the scientific validity of the proposed approach. This is crucial for practical considerations, especially in real-time communication systems.
Response 1:
We appreciate your careful advice. Based on your recommendation, we have revised the earlier manuscript to response reviewers’ comments clearly. In Sec. Conclusions, we mentioned the further research topic. As a result, we could improve the revised manuscript based on your advice and reviewer’s comments.
Point 2: The paper lacks a thorough discussion of potential limitations and challenges associated with the proposed NC-CE methods. Addressing these limitations, even briefly, would contribute to a more balanced and nuanced presentation of the research findings.
Response 2:
We appreciate your careful advice. Based on your recommendation, we have revised the earlier manuscript to response reviewers’ comments clearly. In Sec. Conclusions, we mentioned the further research topic. As a result, we could improve the revised manuscript based on your advice and reviewer’s comments.

Reviewer 3 Report
Comments and Suggestions for Authors
authors proposed a method for noise cancelling-channel estimation for OFDM modulation. to validate the propose method authors show simulation results. the paper is well writing and could be published as it is.
Here is some quotation that authors should take into consideration in preparing the revised version.
Authors proposed a method for noise cancelling channel estimation for OFDM modulation. Simulation results were given to validate the proposed method. However, some concerns must be addressed before the paper could be accepted.
In literature, many estimation method were used fro OFDM, the subject become more commun, is it important to address this point in an original paper.
By using three steps for cestimation, this process will be complex. Authors should clarify this matter by explaining or comparing the complexity degree of the proposed work by other in literature
Many equations were introduced, however, few justification are given, please give more detail about the introduced equations, especially, in section 3.2)
How does authors calculated the PER of the proposed method, is there any equation or it was calculated by simulating the entire transmission chain
Regarding the proposed technique, what was the objective of introducing this cancelation method.
Comments on the Quality of English Language
authors used standart english in the paper
Author Response
Point 1: In literature, many estimation method were used fro OFDM, the subject become more commun, is it important to address this point in an original paper.
Response 1: We appreciate your valuable advice. Based on your recommendation, we have revised the earlier manuscript to response reviewers’ comments clearly. Furthermore, we made our best effort to more clearly describe the proposed methodology in the abstract and introduction section. As a result, we could improve the revised manuscript based on your advice and reviewer’s comments.
Point 2: By using three steps for cestimation, this process will be complex. Authors should clarify this matter by explaining or comparing the complexity degree of the proposed work by other in literature.
Response 2: In this paper, we propose a channel estimation technique applicable to OFDM systems with a small number of pilot symbols, such as IEEE 802.11p. Table 1 shows a complexity comparison with channel estimation techniques applicable to IEEE 802.11p, and it can be seen that the proposed method has relatively low complexity.
We appreciate your valuable advice. Based on your recommendation, we have revised the earlier manuscript to response reviewers’ comments clearly. The typos and grammatical errors have been corrected. Furthermore, we made our best effort to more clearly describe the proposed methodology in the abstract and introduction section.
Point 3: Many equations were introduced, however, few justification are given, please give more detail about the introduced equations, especially, in section 3.2.
Response 3: We appreciate your valuable advice. Based on your recommendation, we have revised the earlier manuscript to response reviewers’ comments clearly. In section 3.2, we explained the differences between this paper and existing papers and added explanations and references to key formulas.
Point 4: How does authors calculated the PER of the proposed method, is there any equation or it was calculated by simulating the entire transmission chain.
Response 4: We appreciate your valuable advice. Based on your recommendation, we have revised the earlier manuscript to response reviewers’ comments clearly.
In section 4, we explained the entire transmission chain used at simulations. Also, we have revised the section 2.2.
Point 5: Regarding the proposed technique, what was the objective of introducing this cancelation method.
Response 1: As mentioned in Sec. Introduction,
In particular, considering that information exchange through V2X communication is very essential for vehicle driving safety, it is necessary to emphasize more accurate channel estimation within the constraints on the pilot subcarrier of IEEE 802.11p [2, 8-12].
In this paper, we incorporate a modified CDP method without a frequency domain reliability test and interpolation, taking into account the CFRs of virtual subcarriers obtained at the previous OFDM symbol time. In addition, the proposed CE method can be implemented as the operation of IFFT/nulling/FFT so as to reduce noise components from the CFRs obtained in the previous estimated CFRs.
Based on your recommendation, we have revised the earlier manuscript to response reviewers’ comments clearly. We made our best effort to more clearly describe the proposed methodology in the abstract and introduction section.

Reviewer 4 Report
Comments and Suggestions for Authors
See the attached file.

Minor editing of English language is required.
Author Response
Point 1: On page 3, in eq. (4), matrix IE has no correct dimensions. Previously it is stated that it is a 52×64 matrix. Therefore, I think this matrix has to be:
Response 1: We appreciate your valuable advice. Based on your comment, we have corrected it.
Point 2: On page 6, row 141: instead of “shown in (11)” −→ “shown in (15)”.
Response 2: We appreciate your valuable advice. Based on your comment, we have corrected it.
Point 3: On page 15, row 289: instead of “875–849” −→ “875–879”.
Response 3: We appreciate your valuable advice. Based on your comment, we have corrected it.
Point 4: Some abbreviations are defined more than once (e.g. CFR, CIR, MSE) or are not defined (e.g. PER, BER).
Response 4: We appreciate your kind advice. Based on your comment, we have corrected all of them.
Point 5: English language has to be revised in some parts (e.g. on page 1, line 31: “very essential”; on page 13, line 221: “From six figures” ).
Response 5: The typos and grammatical errors have been corrected. Furthermore, we made our best effort to more clearly describe the proposed methodology in the abstract and introduction section. As a result, we could improve the revised manuscript based on your advice and reviewer’s comments.

Reviewer 5 Report
Comments and Suggestions for Authors
The authors propose a three-step channel estimation technique for IEEE 802.11p systems. The proposed solution is validated by means of simulations for different channel models. Its performance and computational complexity are compared with the ones of other existing techniques. The authors should take into account the following comments and suggestions in order to improve the overall quality of the paper.
- A column should be added to table 1, in order to clearly specify which is the proposed technique and which are the existing ones (references should be included);
- The figures and tables should be shifted after the text which is commenting them;
Comments regarding grammar/typos:
- …it is difficult to estimate with high precision channel changes… instead of current form (row 29);
- …for obtaining excellent channel… instead of … to have excellent channel (row 62);
- …utilizes the 5.9 GHz band... instead of current form (rows 79-80);
- …express… instead of …expressed… (row before eq. 15);
- …have the following… instead of …have following… (row 151);
- …let us define the mean square… instead of current form (sentence before eq. 20).
- From the six figures… instead of current form (row 221).
Comments on the Quality of English LanguageComments regarding grammar/typos:
- …it is difficult to estimate with high precision channel changes… instead of current form (row 29);
- …for obtaining excellent channel… instead of … to have excellent channel (row 62);
- …utilizes the 5.9 GHz band... instead of current form (rows 79-80);
- …express… instead of …expressed… (row before eq. 15);
- …have the following… instead of …have following… (row 151);
- …let us define the mean square… instead of current form (sentence before eq. 20).
- From the six figures… instead of current form (row 221).
Author Response
Point 1:
- A column should be added to table 1, in order to clearly specify which is the proposed technique and which are the existing ones (references should be included);
- The figures and tables should be shifted after the text which is commenting them;
Response 1: We appreciate your valuable advice. Based on your recommendation, we have revised the earlier manuscript to response reviewers’ comments clearly. In table 1, we add ‘Comments’ column order to clearly specify which is the proposed technique and which are the existing ones. In addition, all pictures were modified to be placed where they were first mentioned.
Point 2: Comments regarding grammar/typos:
- …it is difficult to estimate with high precision channel changes… instead of current form (row 29);
- …for obtaining excellent channel… instead of … to have excellent channel (row 62);
- …utilizes the 5.9 GHz band... instead of current form (rows 79-80);
- …express… instead of …expressed… (row before eq. 15);
- …have the following… instead of …have following… (row 151);
- …let us define the mean square… instead of current form (sentence before eq. 20).
- From the six figures… instead of current form (row 221).
Response 2: The typos and grammatical errors have been corrected. Furthermore, we made our best effort to more clearly describe the proposed methodology in the abstract and introduction section. As a result, we could improve the revised manuscript based on your advice and reviewer’s comments.
